# The Genealogy of Obedience in the Biblical Sources of Jewish Culture

**Bina Nir**

Department of Multidisciplinary Studies, The Academic College of Emek Yezreel, Yezreel Valley 1930600, Israel; binan@yvc.ac.il

**Abstract:** In the past two years, against the backdrop of the Covid-19 epidemic, civil disobedience has been on the rise in Israeli society. While civil disobedience preserves the boundaries of discourse and democracy, it also poses a real danger to the existence of society. Attitudes toward civil disobedience reflect the core values of a particular culture. Thus, attitudes toward obedience or disobedience in Israeli society must be examined with respect to the religious roots of Jewish culture. This article examines attitudes towards the issue of obedience in the formative texts of Jewish culture. Revealing the religious past of the culture enables a re-evaluation of values and stances. To understand the sources of the values relevant to obedience in Israeli society today, I will analyze religious texts using the genealogical method. The subject area of genealogy is the past, yet its purpose is to understand and critique contemporary reality. The main finding is that obedience and disobedience do not appear to be in a dialectical relationship in today's society. Despite the fact that controversy is welcome and prevalent in Judaism, obedience and disobedience do not manifest as contentious or as two sides of a single issue.

**Keywords:** chosenness; civil disobedience; disobedience; genealogy; separateness

## 1. Introduction

As we witnessed the spread of the COVID-19 virus in Israel and the world, we also saw the subject of disobedience and its place in society surface on public and media agendas. The spread of the virus, by all accounts, was attributed to public disobedience in relation to government-mandated safety rules such as lockdowns, prohibition of public gatherings, the wearing of masks, and most recently, vaccinations. On the other hand, disobedience has also been taking the form of demonstrations and protests against the strict regulations, with protesters claiming that, on multiple occasions, the decision to institute a nationwide lockdown was tainted by personal and political motivations. Research about public conduct during the pandemic in Israel has shown a decrease in the level of trust that the public feels toward the resolutions passed by the Government and its institutions (Ayalon 2021; Kimhi et al. 2020).

As with any social and political phenomenon, the current unrest stems from an accumulation of many different factors, all of which have come to a head and erupted violently at this point in time, catalyzed by the pandemic. At the same time, the phenomena of obedience and disobedience in themselves are only partially dependent on specific political and social circumstances and have particular cultural attributes that differ from one people or nation to another. In this article, we shall cast an explorative look at the distant cultural past of obedience and its necessary antithesis, disobedience, in the deep sources of Jewish culture, using the genealogical method. We shall not attempt here to present an extensive genealogy of the subject as a means of comprehending present reality as the history of Jewish culture is too long and varied for such an endeavor. It would require us to trace multiple processes over a far longer timeline than the scope of the present article can encompass. One might even question whether it is even possible to produce a solid

and closed procedural structure around this topic. Thus, we shall at present focus on a limited number of selected cultural patterns pertaining to the subject of obedience while referring to the Jewish biblical canon as a foundational text. While this type of examination will not provide us with a causal link between the deep historical past and the present era, the very existence of these patterns can provide food for critical thought with regard to our culture and its roots.

## 2. The Genealogical Method

Cultures are often based on such deep basic assumptions that they are mostly invisible unless one examines them retrospectively. Genealogy, at its core, allows for a re-examination of values and attitudes (Foucault 1977, p. 152). Our adoption of the genealogical method as a critical practice relies on the ideas of Nietzsche and Foucault: Nietzsche is considered the father of genealogy as a critical method, while Foucault implemented his predecessor's work and expanded its implications (Deleuze 2006, p. 2). Human beings, according to Nietzsche, live historically, meaning that they are aware of and conditioned by their past (Nehamas 1994, pp. 270–72). Hoy (1994) presents Nietzsche as a genealogist convinced that we can never reach the "secret springs" of the world and gain a perfect, all-encompassing perspective on any existing phenomenon. His genealogy can create an empirical environment, but it does not entail a dialectical or logical necessity. Nietzsche rejects the search for the true visage of the world with the argument that the very concept of "the true visage of the world" is moot and nonsensical (Parush 1995, p. 128). "Here it becomes clear", he writes, "how badly man needs . . . in addition to the monumental and antiquarian ways of seeing the past, a third kind, the critical . . . He must have the strength, and use it from time to time, to shatter and dissolve something to enable him to live: this he achieves by dragging it to the bar of judgment, interrogating it meticulously and finally condemning it" (Nietzsche 1980, pp. 21–22). Nietzsche's conception as a genealogist is that of a multitude of perspectives, within each of which the truth is revealed, but none of which is in itself "the truth about the world"—an empty concept to his mind. His central thesis is, therefore, that any claim to truth can only be asserted from a certain perspective.

At the basis of Nietzschean thought resides the psychologist and the evaluator, the writer–thinker pondering customs, thoughts, and behaviors (Blondel 1994, pp. 309–10). His genealogy is a hermeneutic strategy that demands meticulous attention to our historical traces (Conway 1994, pp. 321–28). It is a criticism wherein the critic establishes their own criteria. For example, Nietzsche examines the values of morality and culture based on the standards he presents, such as the will to power (Rusinek 2004, p. 423).

That being said, our preliminary genealogy of the various aspects of obedience will be based mainly on the biblical text, as the Hebrew Bible is a canonical and foundational text for Jewish culture, upon its many factions, as well as Western culture in general (Haran 1995). The creation of the canonical version of the Old Testament probably began in the Hasmonean era with the purpose of forging a national identity that included, among other things, the Hebrew language, and the resulting collection formed the basis for what we now term the Jewish Bible, or the *Tanach* (Davies 2010, p. 372).

Worldviews that are deeply rooted in the religious experience, according to Carl Gustav Jung (1949), often have the hidden potential to be indirectly preserved in secular thought over long periods of time. In our specific case, some of the ideas and values put forward in the Old Testament, as a founding text of Jewish culture, have been preserved over time in the secular manifestations of Jewish culture, all while changing continuously, their roots remaining discernable even across a vast distance of time. William James ([1902] 2002) recognizes three essential components present in all world religions: faith, a super-human order, and the obligation to uphold certain commandments or ordinances. The fulfillment of ordinances is perceived as something that promotes one's connection to the exalted super-human cosmic order. In Judaism, the observance of divine edicts, or *mitzvoth*, is central to the religion and viewed as highly important. The verse "If ye walk in my statutes, and keep my commandments [*mitzvoth*], and do them; then I will give you

rain in due season, and the land shall yield her increase, and the trees of the field shall yield their fruit" (Leviticus. 26:3–4) indicates that God expects the people of Israel to observe the Torah and uphold the *mitzvoth*.[1] According to the doctrine of divine retribution, obedience to the divine decrees is rewarded, while disobedience is punished: "But my people would not hearken to my voice; and Israel would none of me. So I gave them up unto their own hearts' lust: and they walked in their own counsels. Oh that my people had hearkened unto me, and Israel had walked in my ways! I should soon have subdued their enemies and turned my hand against their adversaries" (Psalms. 81:11–14). Obedience is an important and very present value in the Old Testament; however, the same is true of disobedience, as we shall see further on.

When conducting a cultural interpretative analysis of religious texts, one may discover paradoxes arising from the fact that such an analysis does not distinguish between the sacred and the profane, a distinction that is almost trivial from a religious point of view. The divine monotheistic God is all-powerful, all-knowing, and infinite (Klein 2004, pp. 42–43). A cultural understanding of a religious text is therefore problematic as the context that led to the composition and canonization of its words stems from the sacred aspect of the religion. Therefore, the cultural analysis of a religious text is always somewhat limited, as opposed to its interpretation by a believer, as the latter's examination of the text is in search of a subject whose presence is presupposed (Ofir 2000, p. 127).

In this article, we will examine a number of ingrained cultural constructs that are religious in nature and require obedience. Conversely, we shall likewise deeply examine ingrained constructs regarding disobedience, which are, in our view, culturally perceived as conflictual with the act of obedience. Our preliminary genealogy will, therefore, aim to uncover these constructs and the inherent conflict between them. A secondary objective will be to determine whether there is a dialectical relationship between these structures of obedience and disobedience or whether they are not necessarily contradictory structures but rather exist within a comprehensive system of understanding.

### 3. Disobedience from Genesis Onward

For centuries, rulers, kings, priests, and parents presented obedience as a virtue and disobedience as a sin. And yet, Western history, argues Erich Fromm, began with an act of disobedience in the story of the Garden of Eden and, conversely, may end with an act of obedience (Fromm 1981, p. 1). Of course, Fromm does not hold that all disobedience is worthy and all obedience is a sin, but he makes a psychological distinction between a person who can only refuse—whom he calls a rebel rather than a revolutionary because they act out of anger and not out of conviction—on the one hand, and a person who can only obey and never refuses—whom he calls a slave—on the other (Fromm 1981, p. 5). While this subject clearly has psychological aspects, it also involves religious–cultural elements that we shall examine below.

The first and greatest act of disobedience appears, of course, in the story of the Garden of Eden. The tale that culminates with the fall of man is a founding myth in Western society, mainly due to the central status it acquired in the Christian tradition. Man is expelled from Eden because he commits the original sin, the act that is the prototype of all sin to follow, and as a result, human beings are transformed into vulnerable individuals who must strive and compete with the other individuals around them in order to survive (Shoham 2003, pp. 14–17). This one instance of disobedience brings a slew of punishments upon humanity, including: "I will greatly multiply thy sorrow and thy conception; in sorrow thou shalt bring forth children" (Genesis. 3:16); "in the sweat of thy face shalt thou eat bread" (Genesis. 3:19); and the worst punishment of all, "therefore the Lord God sent him forth from the Garden of Eden" (Genesis. 3:23). One alternative approach to the story claims that the gravest infraction consists not of the act of disobeying God's command or eating from the Tree of Knowledge—after all, anyone can make mistakes and break the rules from time to time—but in the failure of the humans to take responsibility for their actions, pointing accusatory fingers elsewhere (Zion 2002, p. 107). Adam lays the blame for the action on

Eve: "And the man said, the woman whom thou gavest to be with me, she gave me of the tree, and I did eat" (Genesis. 3:12), while Eve, in turn, blames the snake: "And the woman said, the serpent beguiled me, and I did eat" (Genesis. 3:13). As if disobedience was not bad enough in itself, the mother and father of humanity refuse to be held accountable for it.

Thus, we can agree with Fromm in arguing that the first human act is an act of rebellion against God's supremacy (Fromm 1966, pp. 21–22). The story of the Garden of Eden, however, can be interpreted from two different and seemingly contradictory perspectives. On the one hand, the story can be read as a pagan tale: God has a tree with magical powers and, driven by jealousy and the desire to retain these powers exclusively to himself, he keeps it from man (Rosenberg 1985, pp. 48–49). This motif is recurrent throughout Greek mythology but is best exemplified by its most well-known example, the myth of Prometheus. Prometheus embodies the character of a rebel who refuses to obey and symbolizes the titanic struggle against the gods. In the series of myths that shaped Western culture, Prometheus symbolizes the liberation of man and humanity (Ohana 2000, p. 3). The Greek myth of Prometheus sees human civilization as based on disobedience and, just like Adam and Eve, Prometheus is punished for his defiance of the divine decree (Fromm 1981, pp. 2–3). The other way of interpreting the story of the fall focuses on God's mercy rather than his jealousy or vengefulness. This foundational myth sees the source of human suffering in knowledge and, as Prometheus too would find out, knowledge is not always a good thing to have: "For God doth know that in the day ye eat thereof, then your eyes shall be opened, and ye shall be as gods, knowing good and evil" (Genesis. 3:5). God's intention is therefore not to prevent mankind from gaining power but to guard his creations from the suffering engendered by knowledge and to preserve the cosmic order, which ends up being disrupted through their act of disobedience.

Hazony (1998), on the other hand, claims the Hebrew Bible to be the only ancient document to promote the idea of disobedience in relation to unjust rules, a principle completely foreign to the pagan world from which it emerged. He argues that while the Old Testament text calls for unquestioning obedience to the Lord God, it does not endorse mindless submission to human authority. When it comes to God, we get countless examples of faith-based obedience even in the direst circumstances, the starkest of which is the binding of Isaac: "And he said, Lay not thine hand upon the lad . . . for now I know that thou fearest God, seeing thou hast not withheld thy son, thine only son from me" (Genesis. 22:12). Disobedience to God, on the other hand, as we have seen in the tale of the fall, is met with unbearable punishment. And yet, Hazony maintains that the Bible also encourages disobedience based on the human conscience, which is instilled in us as the heart of moral independence, for biblical heroes are not ones to submit to orders, even when they come from God and have a tendency to act upon their own moral instincts (Hazony 1998, p. 25). One interesting argument in support of this claim is the very name "Israel" given to Jacob and to the Jewish people as a whole, which means "struggle with God": "Thy name shall be called no more Jacob, but Israel: for as a prince hast thou power with God and with men, and hast prevailed" (Genesis. 32:28). Hazony's reasoning is in line with the distinction made by Fromm between heteronomous obedience to an institution or an authority, which is submission, and obedience to one's conscience, reason and beliefs, which is autonomous obedience, expressive of affirmation rather than submission (Fromm 1981, pp. 4–5).

That said, the degree to which our conscience is truly autonomous is also questionable. Human conscience may be essential for action and growth (Rotenberg 1997, p. 84); however, we often admit to acting in a way we think moral due to the urging of our conscience without this impulse undergoing critical evaluation or reflexive thinking. Of course, as we also know, plenty of atrocities have been committed in the name of the human conscience (Sagi 2010, p. 362). Therefore, even while believing that we are acting out of the "humanistic conscience", which is the inner, autonomous voice of our humanity, we may, in fact, be acting out of an "authoritarian conscience", which is the internalized voice of the authority we wish to please and are afraid to disappoint (Fromm 1981, p. 6). Freud (1961) argues that an excessive number of ordinances and prohibitions and the subsequent development

of guilt through religious institutional mechanisms, social pressure, parental pressure, and the pressure put upon us by our own superego creates a dominant "authoritarian conscience" and is therefore dangerous. By the same token, Freud (1961) warns us against culture, which in many cases, is something that is imposed on a refusing majority by a minority that has managed to seize means of force and coercion. When we are faced with an overabundance of rules, it is not always possible to distinguish between the voice of the "authoritarian conscience" and the voice of the "humanistic conscience".

### 4. "All of Israel Are Responsible One for the Other:" between Obedience and Separateness

Obedience is linked, among other things, to a sense of conformity, cohesion, shared responsibility, belonging, and unity. The actions of the individual, for better or worse, affect the rest of the community. In Judaism, by realizing that they are not separate individuals but part of a group, people feel less lonely and combative and more prone to sympathy for those "others" to whom they in fact belong (Yigal 2012). When a person obeys the precepts of a religion, the rules of a group, or public opinion they feel protected and safe, but, in return, they risk, of course, losing their autonomy (Fromm 1981, p. 8).

"All of Israel are responsible one for the other" is an expression coined by the Jewish sages. In the modern era, it is taken to mean that every Jew is responsible for the wellbeing and welfare of their fellow Jews; however, its original intention is that every Jew bears responsibility for every other Jew's observance of the *mitzvoth*. The phrase comes from the Sifra *midrash* which comments on Leviticus 26:37: "'And they will stumble, one man by his brother': It is not written 'one man because of his brother' (i.e., in running), but 'one man by his brother,' the sin of his brother—whereby we are taught that all of Israel are responsible, one for the other" (Sifra: Bechukotai. 7:5). In his commentary of this verse, Rashi repeats this sentiment, in slightly different wording: "A Midrashic explanation is: one will stumble on account of the other, for all Israelites are held responsible for one another." Being responsible for one another conveys a strong sense of unity and accountability to one's peers: an individual who disobeys the *mitzvoth* brings woe onto others.

The Old Testament too contains a collective aspect of retribution, alongside the individual retribution previously discussed. This collective aspect sees the deeds of the individual deciding the fate of Israel as a whole, especially when that individual has an exceptional status, such as a ruler or a king, for instance. There are also instances when the entire people are punished due to the actions of one in their midst (Weiss 1987), as the individual has the potential to lead the entire group astray: "If you hear it said . . . that troublemakers have arisen among you and have led the people of their town astray, saying, 'Let us go and worship other gods' (gods you have not known), then you must inquire, probe and investigate it thoroughly. And if it is true and it has been proved that this detestable thing has been done among you, you must certainly put to the sword all who live in that town . . . Then the Lord will turn from his fierce anger, will show you mercy, and will have compassion on you" (Deuteronomy. 13:12–17).

There is, however, in Judaism a deep-seated construct that, to my mind, often finds itself in inherent conflict with the principle of obedience to group and community rules out of mutual accountability and that is the sense of separateness. On the one hand, Jewish culture ordains collective responsibility and love of the other, yet, on the other hand, it places great emphasis on the separation between "us" and "them". The Jewish sources contain hierarchic orders of peoples and nations, as well as internal hierarchies between different Jewish tribes and factions. It is, therefore, easy to see how certain groups might be perceived as "other" and "different" and become excluded from the principles of mutual accountability and shared community.

The famous decree "love thy neighbor as thyself" (Leviticus. 19:18) is a very important notion in the Bible, what Rabbi Akiva considers "the greatest principle of the Torah" (Jerusalem Talmud: Nedarim. 30b). Man is commanded to love others as he loves himself. Ostensibly there can be no greater unity and mutual respect, which is why many of the

social rules of behavior we follow to this day are based on this one principle. In Christianity too, love of one's neighbor is a central concept. The verse is presented by Jesus and by Saint Paul as the basis of Christian morality: "And Jesus answered him, The first of all the commandments is, Hear, O Israel; The Lord our God is one Lord . . . And the second is like, namely this, Thou shalt love thy neighbor as thyself. There is none other commandment greater than these" (Mark. 12:29–31).

And yet, a closer look at the full verse, as it appears in Leviticus—"Thou shalt not avenge, nor bear any grudge against the children of thy people, but thou shalt love thy neighbor as thyself: I am the Lord" (Leviticus. 19:18)—tells us that the commandment to love "thy neighbor" actually refers to members of "thy people". This is not an ordinance of universal love but of the love of Israel, commanding the people of Israel to love one another. As Maimonides decrees: "It is a positive commandment for each man to love each Israelite as himself, as the verse says: 'Thy shalt love thy neighbor as thyself'" (Maimonides 1983, p. 44). According to this cultural conception, there can be no identification, merging, acceptance, or unity with the "other".

As mentioned above, the Old Testament presents a clear and distinct hierarchy between peoples, groups, and even individuals. A relatively significant portion of the biblical discourse is allotted to ranking, separating, and distinguishing (Nir 2016). This very deep-seated element in Judaism can often go against the principle of mutual accountability and obedience for the sake of the group. The idea of "the chosen people", for instance, appears many times in the biblical text: "I give waters in the wilderness, and rivers in the desert, to give drink to my people, my chosen" (Isaiah. 43:20); "the Lord thy God hath chosen thee to be a special people unto himself, above all people that are upon the face of the earth" (Deuteronomy. 7:6); "it is he that hath made us, and not we ourselves; we are his people" (Psalms. 100:3). The concept of a "chosen people" is, of course, diametrically opposed to the idea of a universal God. At the very basis of the notion of being "chosen" is the idea of being singled out from the others, of separateness and comparison.

The idea of being "chosen" in the Bible, as previously mentioned, is not exclusive to a people but can also apply to tribes or groups. Among the twelve Israelite tribes, God sets the Levi tribe apart from the others: "And I, behold, I have taken the Levites from among the children of Israel . . . therefore the Levites shall be mine" (Numbers. 3:12). Out of the Levi tribe, God selects the *Cohanim* (the priestly caste): "Take the sum of the sons of Kohath from among the sons of Levi" (Numbers. 4:2). Aharon the priest, who is the progenitor of all the priests of Israel, is the son of Amram who is the son of Kohath, and Kohath is the second son of Levi, son of Jacob. Their chosen status bestows onto them the duty "to do the work in the tabernacle of the congregation. This shall be the service of the sons of Kohath in the tabernacle of the congregation about the most holy things" (Numbers. 4:3–4).

The selection of the chosen people from among all other peoples and of one tribe from among the twelve can form the cultural basis for a comparative perception of various groups: a tribal, sectarian, differentiating and non-unifying outlook. Obedience to the idea that "all of Israel are responsible one for the other" remains an important value in Judaism but mostly when it comes to the *mitzvoth* between man and God. In the *mitzvoth* between man and his peers, this value is only salient within the boundaries of the peer group, and, therefore, it may lose its broad societal and universal impact. This, to my mind, is a built-in conflict that does not necessarily engender a dialectical relationship between internal obedience within the community to which we belong, out of mutual accountability and the external disobedience toward the "others", as those within the community might perceive those outside of it.

## 5. "We Will Do and Be Obedient" between Obedience and a "Stiff-Necked People"

In the biblical account of the reception of the Torah on Mount Sinai, the Israelites are quoted as saying only that they will do as the Lord says: "And Moses came and called for the elders of the people, and laid before their faces all these words which the Lord commanded him. And all the people answered together, and said, All that the Lord hath spoken we will

do" (Exodus. 19:7–8). The combination of "we will do and be obedient" appears only later: "And he took the book of the covenant, and read in the audience of the people: and they said, All that the Lord hath said will we do, and be obedient" (Exodus. 24:7). The accepted interpretation of the verse sees this phrase as an expression of the symbiotic relationship between the recognition of the burden of the *mitzvoth* and the recognition of the Divine Power. In the writings of the sages, the expression "we will do and be obedient" becomes a distinct symbol of faith-based obedience that signifies an unconditional acceptance of the Torah: "Rabbi Simai taught: When Israel accorded precedence to the declaration 'We will do' over the declaration 'We will obey,' 600,000 ministering angels came and tied two crowns to each and every member of the Jewish people, one corresponding to 'We will do' and one corresponding to 'We will obey'" (Babylonian Talmud: Shabbat. 88a). There is, however, another interpretation which states that God had forced the people of Israel to accept the burden of the Divine Power out of apprehension that they would be too terrified by the revelation on Mount Sinai: "The Almighty held the mountain over them like a barrel—even though they had already said 'We will do and we will obey,' perhaps they retracted when they saw the great fire [on the mountain] that caused their souls to depart" (Babylonian Talmud: Tosafot Shabbat. 88a:5).

Absolute obedience and the unconditional acceptance of the Torah are also implicated in the doctrine of divine retribution. Jewish ethics are based on a reward/punishment dialectic between man, the community, and God. The nation and the individual determine their fate by their actions (Yakobson 1959). The designation of Israel as the chosen people confers upon them the duty to serve as an example to all other nations, and, therefore, they are punished severely for any infraction against God's decrees and *mitzvoth*: "You only have I known of all the families of the earth: therefore I will punish you for all your iniquities" (Amos. 3:2). In the divine court of judgment, a man is rewarded and punished not only based on his own actions but based on those of his fathers. Furthermore, the repercussions do not stop there, for the whole nation is implicated in the act of sin. Roughly three thousand people participated in the fabrication of the golden calf (Exodus. 32:28) and yet the entire nation of Israel is punished for it: "And the Lord plagued the people because they made the calf, which Aaron made" (Exodus. 32:35). Just as obedience and strict observance of the *mitzvoth* come with the promise of palpable reward, disobedience entails truly dreadful penalties:

> If thou wilt not hearken unto the voice of the Lord thy God, to observe to do all his commandments and his statutes . . . all these curses shall come upon thee, and overtake thee: cursed shalt thou be in the city, and cursed shalt thou be in the field . . . cursed shall be the fruit of thy body, and the fruit of thy land . . . and thou shalt not prosper in thy ways: and thou shalt be only oppressed and spoiled evermore, and no man shall save thee" (Deuteronomy. 28:15–29)

The absolute obedience inferred from "we will do and be obedient" is at odds with the biblical description of the people as "stiff-necked" or obstinate: "And the Lord said unto Moses, I have seen this people, and behold, it is a stiff-necked people" (Exodus. 32:9); "they would not hear, but hardened their necks, like to the neck of their fathers, that did not believe in the Lord their God" (2 Kings. 17:14). The biblical story reveals that it is because the chosen people are "stiff-necked" that they have a hard time obeying the Lord their God, as well as their own leaders. In the Book of Numbers, the people disobey the laws of the Hebrew God and choose Moabite gods and women over him: "And Israel abode in Shittim, and the people began to commit whoredom with the daughters of Moab. And they called the people unto the sacrifices of their gods: and the people did eat, and bowed down to their gods" (Numbers. 25:1). In the Book of Judges, the people once again disobey and sin by worshipping foreign gods: "And the children of Israel did evil again in the sight of the Lord and served Baalim, and Ashtaroth, and the gods of Syria, and the gods of Zidon, and the gods of Moab, and the gods of the children of Ammon, and the gods of the Philistines, and forsook the Lord, and served not him" (Judges. 10:6).

The people are not the only ones depicted as "stiff-necked" and disobedient, failing time and time again to fulfill their covenant with God. Their leaders and kings, the chosen among the chosen people, also find it hard to follow directions, as the Bible shows on countless occasions and do evil in the sight of the Lord. King David fails to obey the tenth commandment: "thou shalt not covet thy neighbor's wife" (Exodus. 20:17). King Solomon goes against the rules God lays out for the future kings of Israel: "Neither shall he multiply wives to himself, that his heart turn not away: neither shall he greatly multiply to himself silver and gold" (Deuteronomy. 17:17). Jeroboam sins by worshipping false gods: "Whereupon the king took counsel, and made two calves of gold, and said unto them, It is too much for you to go up to Jerusalem: behold thy gods, O Israel, which brought thee up out of the land of Egypt . . . And this thing became a sin" (1 Kings. 12:28–30). In fact, there is a whole dynasty of rulers described in the First Book of Kings who do evil in the sight of God:

> And Nadab the son of Jeroboam began to reign over Israel . . . And he did evil in the sight of the Lord, and walked in the way of his father, and in his sin wherewith he made Israel to sin . . . In the third year of Asa king of Judah began Baasha the son of Ahijah to reign over all Israel . . . And he did evil in the sight of the Lord, and walked in the way of Jeroboam, and in his sin wherewith he made Israel to sin. (1 Kings. 15:25–34)

In the Second Book of Kings, we find a description of King Zedekiah's disobedience: "Zedekiah was twenty and one years old when he began to reign . . . And he did that which was evil in the sight of the Lord . . . For through the anger of the Lord it came to pass in Jerusalem and Judah, until he had cast them out from his presence, that Zedekiah rebelled against the king of Babylon" (2 Kings. 24:18–20). The punishment for a particular king's acts of disobedience happens to be the destruction of the First Temple. As we can see, disobedience seems to be a dominant motif in the lives of the kings of Israel, one that characterizes the entire period of the ancient kingdom of Israel. The fall of the kingdom is therefore described as a direct consequence of the failure on the part of the kings to accept the limitations placed on their power and authority by the Jewish tradition.

In his dialogical interpretation of the biblical text regarding the disobedience of leaders, the Jewish philosopher Martin Buber argues that the Bible came to teach us that the path of truth lies not in the realm of achievement but in the depth of human failure (Buber 2002, p. 36). The biblical author presents repeated acts of disobedience and sins to emphasize the gap between man and God. Looking at the text through the eyes of cultural interpretation which sees the text as a constitutive literary work, we find that disobedience to God, as expressed in the many stories involving leaders and rulers, is not revolutionary, in the sense that it does not attempt to produce change or create something new. It is simply a recurring motif of rebellion derived from the failure to gain ultimate authority, which is in conflict with the concept of "we will do and be obedient".

## 6. Obedience Driven by the "Authoritarian Conscience"

As previously mentioned, Fromm distinguishes between obedience driven by the "authoritarian conscience" that is the internalized voice of authority and obedience driven by the "humanistic conscience" that is the internal, autonomous, and authentic voice that calls us back to our humanity (Fromm 1981, p. 6). Though the ordinance itself may be humanist in nature, such as charity, for example, the desire to fulfill it can still be driven by the "authoritarian conscience", thereby preventing the person from exercising their reflexive awareness and seeking out the deeper meaning behind the act. Freud (1961), as previously mentioned, also believed that excessive obedience out of the "authoritarian conscience" may distance us from humanistic reason and autonomy of thought.

In Judaism, on the other hand, though intention remains important, there is greater attention paid to the external appearance of conduct to the visible manifestation of one's obedience. For example, when the Talmud defines the term "*talmid chakham*" ("Torah scholar"), it describes the clothing of one pretending to this title, among other attributes:

"And Rabbi Ḥiyya bar Abba said that Rabbi Yoḥanan said: a Torah scholar on whose clothes a fat stain is found is liable to receive the death penalty" (Babylonian Talmud: Shabbat. 114a). The main reason for this Halakha is that dirty clothing causes blasphemy. However, there is also a broader interpretation that states that the visible stain on the garment is an external projection of the *talmid chakham*'s inner state. The Jewish Halakha also promotes the concept of *mar'it ayin* (literally translated as "appearances") which prohibits actions that are not sinful in themselves but may appear to be sinful. The source of the prohibition is found in the Babylonian Talmud, where there is a prohibition on consuming fish blood even though there is no explicit injunction against it in the Torah as fish are not considered meat. The ban is based entirely on appearances, as onlookers may think that it is the forbidden blood of a land animal: "Rav says: Fish blood that one collected in a receptacle is prohibited for consumption because it would look as though one is consuming the blood of an animal or bird" (Babylonian Talmud: Keritot. 21b).

The ancient Jewish view judged a person according to their deeds only. On the matter of intent, there was a dispute between the Hillel and the Shammai schools of thought. The House of Shammai maintained that a person should be evaluated based on their actions. By contrast, the House of Hillel emphasized the value of intention as a factor in assessing one's virtue. An example that clarifies the difference between the two houses on the subject of intention can be seen in the Keritot tractate of the Mishnah: "Rabbi Judah said: even if he intended to pick figs and he picked grapes, or grapes and he picked figs, white [grapes] and he picked black ones, or black and he picked white ones Rabbi Eliezer declares him liable to a hatat. And Rabbi Joshua declares him exempt" (Mishnah: Keritot. 4:3). In this passage, there is a dispute between Rabbi Joshua Ben Hananiah of House Hillel and Rabbi Eliezer Ben Hurkanus of House Shammai regarding the fitting judgment for a man who meant to pick grapes on the Sabbath but picked figs—an action forbidden by the Torah—or vice versa. Rabbi Eliezer sees the deed—the infraction of a law stipulated in the Torah—as the main reason to find the man guilty. On the other hand, Rabbi Joshua exempts the man from punishment as he considers the man's guilt and subsequent punishment based on his intention, the intention being the main criterion on which the judgment must be based.

The matter of intention and obedience based on the humanistic as opposed to the authoritarian conscience is evoked in the context of confession on Yom Kippur. In Judaism, confession must be general and exclude names, times, and places. The confession is made in the first-person plural, without pointing to the transgressive individual: "We have trespassed; we have betrayed; we have stolen; we have slandered; we have caused others to sin; we have caused others to commit sins for which they are called wicked" (Machzor Yom Kippur Ashkenaz Linear 2020, The Morning Prayers, Amidah). The plea for forgiveness is likewise collective: "And so may it be Your will . . . that You pardon us for all our careless sins, and that You forgive us for all our deliberate sins, and that You grant us atonement for all our rebellious sins" (Machzor Yom Kippur Ashkenaz Linear 2020, The Morning Prayers, Amidah). However, it is also very detailed so as to include everyone: "for the sin we committed before You with an utterance of the lips . . . with knowledge and with deceit . . . by improper thoughts . . . by joining in a lewd gathering . . . by desecrating the divine Name . . . by cheating a fellow-man" etc. (Machzor Yom Kippur Ashkenaz Linear 2020, The Morning Prayers, Amidah). Confession is therefore not personal but collective and spoken aloud in front of the congregation.

There is no private sin that is not part of the collective moral state, and there is no moment in the condition of the collective that is not affected by the sins of the individuals of whom it is comprised (Ofir 2000, p. 134). In Judaism, the personal confession is whispered internally, without sharing it with the public, on all days of the year and before one's death (Cassuto 1973). In terms of confessing the details of the sin, there is some disagreement on the matter: "Some rule that one must name the specifics of the sin . . . for the sake of shame . . . so that the sinner is ashamed of his sins . . . and some are of the opinion . . . that the specifics of the sin need not be disclosed, he can speak the sin in the alphabetical order, even out loud, for this is not specific since everyone speaks it equally" (Zevin 1965, pp. 412–55). When there

is one day in the year dedicated specifically to atonement and when the confession of sins is done publicly in the first-person plural according to a readymade list of sins, we might consider this an impingement on the personal responsibility required to atone for misdoings internally with intention.

Intention is no less important when it comes to acts of disobedience. In the humanistic sense, Fromm (1981) maintains, disobedience is an action aimed not against something but for something, most often for one's right to see, to say what it is they see, and to refuse to say that which they do not see. Such a form of disobedience is existentialist at its core as it is performed out of choice and responsibility (Fromm 1981, p. 24). In this context, Fromm describes the prophets who remained unimpressed by might in the sense of a ruling power and spoke the truth even if it meant disobedience, imprisonment, banishment, or death. They reacted because they felt a responsibility to do so (ibid., 15). In the words of the prophet Amos, "The lion hath roared, who will not fear? the Lord God hath spoken, who can but prophesy?" (Amos. 3:8).

## 7. Conclusions

According to Nietzsche's (1980) genealogical method, the observation of the past is an essential part of being human, as individuals and as a society. The preliminary cultural genealogy explored in the present article, which looks at the biblical past of Jewish culture in relation to the various aspects of obedience, does not strive to teach us about the present in any causal or systematic context. Its purpose is to shed light on the subject and link it to the present by simply exposing it. This kind of genealogy allows us to learn about the roots of our culture across the distance of time by exposing certain constructs that are positioned in a structural conflict.

Biblical law includes civil and criminal laws, moral and social laws, religious and ritual laws, all embedded within historical narratives. The biblical text contains many laws that constitute the central values of Judaism. Obedience to these laws is highly rewarded, while disobedience entails harsh punishment (Weiss 1987). Nevertheless, as we have seen, the biblical narrative is replete with stories of disobedience, from Genesis onward. The message that emerges from the narrative sequence, whether it takes place in the Land of Israel or in exile, is that human laws are only binding when they are just (Hazony 1998, p. 34), and any person can break the boundaries of the law.

As previously mentioned, Fromm (1981) distinguishes between the authoritative conscience, which is the internalized voice of external authority—the consciousness that develops man's obedience, and the humanistic conscience championed by Fromm, which is an intuitive voice humans inherently possess, one that helps us decide what is human and what is not. According to Fromm, it is easier to act out of the authoritative conscience as opposed to the humanistic conscience (Fromm 1981, p. 51). Moreover, the inner conscience tends to speak to us indirectly and covertly, so that often we are not even aware of hearing the voice of our conscience. The two kinds of conscience—authoritative and humanistic—are present in each and every one of us, sometimes juxtaposing or even blending with each other, and it is up to the individual person to observe the power balance between the two and make choices accordingly.

The father of logotherapy, Victor Frankl (2014) also defines the humanistic conscience as man's inherent capacity to reveal the significance of a given situation. Besides being intuitive, this conscience is active and, at times, may direct a person to do something that goes against the prescriptions of society. Our conscience has the power to reveal the meanings that conflict with accepted values, and according to Frankl, in such instances, one must act according to one's conscience (Frankl 2014, pp. 37–42). Similarly to Fromm and Frankl, Rabbi Schulweis (2010) claims that one can view the conscience as an inner compass guiding our lives, a compass that we must seek out and look to time and time again in resistance to the oppressive culture of unquestioning obedience. No wonder that the Biblical Hebrew word for "conscience," *matzpun*, comes from the verb *tzafan*, "to hide," referring to the hidden part of our nature that orients our decisions.

In Jewish culture, while authoritative obedience is a supreme value, as derived from the "we will do and be obedient" verse, humanistic disobedience remains present and dominant. Obedience is a supreme value in reference to the laws of the Torah given to Moses as divine law. Disobedience, on the other hand, is sanctioned in the context of human laws and regimes. As we have seen in the biblical story, disobedience crosses all strata of the people: individuals, families, judges and kings. The latter fail to obey the divine law and their punishment is therefore severe.

Disobedience is an important component of democratic existence as blind obedience can pose a danger. In this context, we have also seen the inherent conflict created when the obedience described in the phrase "all of Israel are responsible one for the other" remains limited to one's peers, as argued by Maimonides. Of course, not all disobedience is a virtue, and not all obedience is a vice. Such a worldview ignores the dialectical relationship between obedience and disobedience (Fromm 1981, p. 4). The biblical narratives we have examined in the confines of the present articles do not always attest to a dialectical relationship between obedience and disobedience; they are not always necessarily in conflict and are not depicted as opposite sides of the same coin. Disagreements in Judaism are a desirable and common thing: "every dispute that is for the sake of Heaven, is destined to endure" (Mishnah: Pirkey Avot. 5:17); dialectical understanding has the role of presenting two contradictory sides within one great truth. Both obedience and disobedience out of belief, ideology, or internal and authentic moral rules are important to the existence of a free society as a society, in general, tends to "normalize" its members (Arendt [1958] 2013, p. 40). Perpetual rebellion and nonacceptance of authority as a way of life or, alternatively, blind obedience are equally a danger; therefore, the dialectic between obedience and disobedience is important. Therefore, the conclusion that emerges here is that a culture in which authoritarian obedience to divine commandments is a central, important, and sublime value may find itself rebelling against human authority because these are two separate systems of authority that are neither opposed nor correlated to each other.

**Funding:** This research received no external funding.

**Data Availability Statement:** This study did not report any data.

**Conflicts of Interest:** The author declares no conflict of interest.

## Note

[1] The analysis presented in this paper is based on the King James Version of the Bible as I feel it most accurately reflects the nuances of the original Hebrew.

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
