# Peer review of "The Genealogy of Obedience in the Biblical Sources of Jewish Culture"

_genealogy, doi:10.3390/genealogy6040084_

Round 1

Reviewer 1 Report (Previous Reviewer 1)

I wish to thank the authors for clearing the notion of the genealogical method, and I apologize for misunderstanding it earlier. From my point of view, the method and general goal of the article are now eloquently explained, which also makes the conclusion much clearer. Still, in my opinion, even though the paper is built around Nietzsche’s conception of genealogy, the authors would do well to refer their readers to relevant studies in the field of the Hebrew Bible. Suppose they wish to go down that road. In that case, they can start by looking at studies such as Joseph Ryan Kelly’s “Orders of Discourse and the Function of Obedience in the Hebrew Bible,” The Journal of Theological Studies, Volume 64, Issue 1, April 2013, Pages 1–24, https://doi.org/10.1093/jts/flt016; or with the references listed at the end of encyclopedic entries such as Anne W. Stewart’s “Moral Agency in the Hebrew Bible,” Oxford Research Encyclopedia of Religion.

But I understand it may not be necessary. To me, it boils down to a question of genealogy as a discipline. If it is “a criticism wherein the critic establishes their own criteria” (lines 75–76), then perhaps there is no need to burden the inquiry with discussions conducted under different criteria. I thus prefer to deflect the final decision to the editors, who are indeed more qualified than I do in the field of genealogy.

Unfortunately, the authors focused mainly on my review’s first and major point and did not address the relatively minor one. Indeed, explaining the goal and methodology of the paper went a long way in establishing its structure. But there is still a need to explain the different parts of the article and how they contribute to the main argument. That, in addition to transitions between sections, will help readers follow the argument and can improve the article’s coherence and cohesiveness. Please see my original review for examples and paragraphs that I find unclear or incoherent. If such constraints, helpful as I believe they can be, impair the nature of the inquiry, maybe they are not necessary.

Author Response

Response to Reader 1

Thank you for your response to my corrections. Your insightful comments in the first round focused on the topic of genealogy and helped me understand that I had not made sufficiently clarified that concept methodologically and conceptually and that I need to offer a deeper and more precise treatment of it.

With regard to the other comments, including the complaint that I made use of contradictory approaches, like that of Freud on the distinction between social pressure and the autonomous “humanistic conscience,” I explicitly state that “That said, the degree to which our conscience is truly autonomous is also questionable. Human conscience may be essential for action and growth (Rotenberg 1997: 84), however, we often admit to acting in a way we think moral due to the urging of our conscience without this impulse undergoing critical evaluation or reflexive thinking. And of course, as we know, plenty of atrocities have been committed in the name of the human conscience (Sagi 2010: 362).”

The human conscience is elusive – people often claim to have acted morally and followed their conscience while experiencing themselves as compelled to act in that way without their having reflected on and subjected that compulsion to critique. Moreover, it goes without saying that wrongs have often been perpetrated as acts of conscience.

My response is similar concerning the unresolved contradiction, as it were, between the perspectives of Rosenberg and Hazony. One of the central claims of the article is that there exist understandable conflicts in biblical sources concerning obedience and disobedience. My presentation of them as conflicting contributes to the central argument.

Reviewer 2 Report (Previous Reviewer 2)

I can see some improvement from the first draft to this draft, but I remain unconvinced that this article is worthy of publication at this time.  There remain a number of underlying assumptions that elide real disputes in contemporary Israeli society and, related, Jewish life, and I see little in the manuscript to tie the deeper examination of Jewish religious texts to the apparent rise in civil disobedience in Israeli society in light of the pandemic.  This is, perhaps, a different article if it is framed simply as being about the Hebrew Biblical origins of civil disobedience -- which, in fact, appears to be the true content of the manuscript -- rather than tying these origins to contemporary Israeli society against the backdrop of the pandemic.  I do not see anything in this draft that makes that particular connection.

That is the bottom line for me:  if you make this an article about the Hebrew Biblical roots of civil disobedience, I think you have something here worth publishing.  If you are going to tie those roots to something going on in contemporary Israeli society in light of the pandemic, you have to illustrate that relationship but you have not done so yet.

Author Response

Reader 2

In response to the excellent comments you made in the first round, I have now made clearer in the article (as was always my intention as I mentioned in my first response) that the article offers a preliminary genealogy of the biblical roots of civil disobedience. This article in no way seeks to connect those biblical roots to civil disobedience that arose in Israeli society during the pandemic or today. The first few lines of the introduction merely mention the pandemic as the context in which civil disobedience has become part of popular discourse. This inspired my curiosity to investigate the concept in terms of the foundations of Jewish culture. I never claimed that there is a causal connection between the cultural past and the cultural present and I believe I made this clear in the second version of the article that includes corrections in response to your helpful comments, which revealed to me that this point was not clear enough in the original.

With thanks,

Round 2

Reviewer 1 Report (Previous Reviewer 1)

I appreciate and satisfy with the author's reply to my comments and recommend the article for publication. It will make a great contribution to Genealogy.

Reviewer 2 Report (Previous Reviewer 2)

No further comments.

This manuscript is a resubmission of an earlier submission. The following is a list of the peer review reports and author responses from that submission.

Round 1

Reviewer 1 Report

The paper is thought-provoking, well written, and exemplifies excellent erudition. However, I am afraid it cannot be considered a work of academic scholarship. It fails to achieve its unrealistic claim and lacks a coherent methodology and a consistent narrative that would make it fact-based research. Thus, it read as a creative contemplation, not uninteresting, but unsuitable for an academic journal.

First and foremost, the paper lacks a clear and feasible goal and a clear conclusion. If its purpose is to track the cultural sources of Israeli disobedience, it falls about 1,500 years short. It deals mainly with biblical and Talmudic literature and presumes it amounts to a new understanding of modern-day Israeli society without providing more than a few psychological references to mitigate this disparity between different times and cultural spheres. If the authors wish to present a complete genealogy of obedience and disobedience in Judaism, from the Bible to present-day Israel, they would have to examine the entirety of Jewish history and what role different attitudes played in different times. However, that seemed far beyond the scope of a single article. Thus, I recommend that the authors opt for a more modest goal, choose the appropriate field, and devise the methodological tools accordingly. Suppose their goal is to explore the sources of obedience and disobedience in modern-day Israeli society. In that case, they will do well to turn to social sciences and use appropriate tools and methodologies. If they wish to present a work of biblical scholarship, they should focus on that venue and provide more references to other works in the field; the same can be said regarding ancient rabbinic literature. 

The lack of a clear goal and focus also leads to methodological ambiguity and inconsistency. The paper fails to provide a true genealogy that examines historical processes. Instead, it presents an ahistorical narrative shielded from changes in time and space by Jungian psychoanalysis. From time to time, the authors use “cultural analysis” without explaining why they chose the methodology and to what extent. Had the paper presented a clear and consistent methodological approach and a specific field of study, it might be able to explain how its authors chose to rely on the works referenced throughout the article while others are overlooked. Without such basic coordinates, it seems the authors pick whichever study fits their narrative.

With regards to said narrative, the authors would do well to present a clear argument they wish to put forward and add a preface explaining the different parts of the article and how they contribute to that argument. When they will, I believe they will notice several problems, including conflicting perspectives as well as unanswered difficulties, that the article is pointing out without resolving. For example: in lines 184–192, Freud is brought to challenge the distinction between social pressure to supposedly autonomic “humanistic conscience,” but the authors do not resolve this difficulty and later (in lines 378–386, for example) continue to rely on the distinction between authoritarian and authoritarian conscience. Similarly, the authors overlook the contradiction latent in Rosenberg’s and Hazony’s perspectives (lines 135–175), where one brings forward a Greek myth of disobedience to God. At the same time, the other claim there is no such phenomenon in non-Jewish ancient literature. These and other discrepancies, including the lack of coherence between the opening and the conclusion, contribute to a fragmented paper that leaves the reader confused.

It is with a heavy heart that I recommend rejecting this paper. It will be an important contribution if the authors wish to focus on biblical scholarship and provide a historical analysis of obedience and disobedience tendencies throughout scripture. But that would require a much better acquaintance with the current state of the field in order to build a proper foundation on which psychoanalysis can be added.

Reviewer 2 Report

There is a nugget of an interesting idea here that I would like to see published, but it needs to be better developed.  Your introductory premise, that an apparent increase in civil disobedience in Israel has roots in ancient Jewish texts illustrating a religious imperative to obey God’s commandments and a history of legal disputation around the meaning of those commandments and how they are to be obeyed, has merit.  However, it appears to be a device solely to introduce the analysis you want to present, which need not necessarily have anything to do with civil disobedience during the pandemic.  Similarly, the analysis itself appears to make several simplifying assumptions about contemporary Israeli Jews, minimizing key aspects of diversity that have bearing on the degree to which different kinds of Israeli Jews are likely to defer to the authority of the ancient texts or the traditional values they convey.  If the article is truly about tracing the roots of contemporary civil disobedience in Israel to these ancient texts, that’s a connection that not only has to be spelled out more clearly, it needs to be explored in sufficient depth to tie the premise to lived reality.

The following notes do not necessarily call for revisions in and of themselves, but some of them illustrate the issues I highlighted in the paragraph above:

Line 47, reference to “Old Testament”:  Genealogy is not my standard methodology so please forgive me if this is the accepted parlance and I am simply nit-picking, but the Jewish community, by and large, does not refer to their sacred texts as the “Old Testament.”  This is a term that is often interpreted by Jews to imply (or outright claim) supersession by the Christian “New Testament,” which is not sacred for Jews.  Jews instead refer to the 24 books in question using the terms you employ a few lines later.

Lines 67-69, on James’ conception of necessary components of religion:  In the context of Israelis, these components are problematic.  To wit, the Pew Research Center’s 2016 survey of Israeli society revealed that only half of Israeli Jewish adults believe in God with absolute certainty, 64% are opposed to halacha (Jewish religious law) being state law for Jews, and although the Orthodox form of Judaism that requires strict observance of mitzvoth is perhaps the only broadly recognized form of Judaism in Israel (and certainly the only officially recognized version), it is a form that is simply not practiced by the vast majority of Israeli Jews.  In practice, many Israeli Jews reflect an old joke that “the synagogue [they] don’t attend is Orthodox,” and others identify with the Masorti or Progressive movements, which have somewhat different approaches to theology, religious practice, and Jewish peoplehood.  One can be Jewish and be accepted as a Jew without faith, belief in a super-human order, or any real sense of obligation to uphold mitzvoth.  Some may sneer at the Jews who do not fulfill these criteria, or dismiss them as “bad Jews,” but they are still generally accepted as Jews.

Lines 88-92, conceptions of religion and God:  I find this conceptualization insufficient when dealing with Jewish populations.  Judaism as a religion compels faith in an omnipresent, omnipotent, and omniscient God; Judaism as an ethno-national category includes many people who lack this faith altogether, some of whom nevertheless observe halacha and are thus simultaneously faithless and faithful.  That’s part of what makes studying Jews so complicated – the cultural context you are emphasizing remains critical. Still, its meaning is not necessarily consistent between the faithless and the faithful in ways that are not as simple as for most other faith groups.

Lines 155-175, on disobedience to unjust rules:  The key here seems to be who gets to decide which rules are just and which are unjust, and thus who has the right to wield authority and whose authority should be resisted.

Lines 176-192, on conscience:  This sounds an awful lot like literature from social psychology on the impact of culture.  For Durkheim, whose work you have previously cited in this paper, the human psyche is a product of the cultural and social milieu in which it develops and functions.  For Mead, attending to and incorporating the views of others into the self is an ongoing, iterative process requiring constant calibration as shared understandings within society develop.  But it also suggests that people with different reference points – say, Haredi vs. Dati Leumi vs. Masorti vs. Progressive vs. generic chiloni – are going to have very different collective consciences and understandings of which rules are just vs. unjust and which authorities deserve deference and which should be disobeyed.

Lines 202-213, on “all of Israel are responsible for one another”:  There are also texts that suggest that one Israelite may be directly responsible for leading another astray (e.g., Deuteronomy 13).

Lines 303-317, on agency, obedience, and justice:  One wonders whether the people truly have the freedom to determine their fate by their own actions if God already knows how they will act, and similarly, whether obedience in the face of the prospect of Divine retribution truly reflects the freedom to choose one’s fate or simply coercion.  And although one may justify the punishment of all of Israel for the sins of the 3,000 who constructed the Golden Calf by the previously cited maxim that “all of Israel are responsible for one another,” one may also question whether it is just to punish the not-yet-born descendants of those who were present but did not prevent the sin.  In the context of which rules and authorities deserve obedience and which ought to be disobeyed as seen from the perspective of multiple social locations within contemporary Israeli Jewish life, these seem like important questions.

Lines 506-510, on Rawls’ conception of civil disobedience:  What about when the majority is significantly abridging the basic civil rights of a particular minority?  Many Israelis feel that the state routinely violates the civil rights of its Palestinian minority.  Similarly, members of the Masorti and Progressive movements feel discriminated against under Israeli law, both in terms of support for their institutions and in regard to matters of personal status.  Can civil disobedience be appropriate even when directed against the majority?